# The Side-Effects of the COVID-19 Pandemic: Increased BMI z-Score in Children with Overweight and Obesity in a Personalised Lifestyle Intervention One Year after the Start of the Pandemic in The Netherlands

**DOI:** 10.3390/nu14091942

**Published:** 2022-05-05

**Authors:** Lisanne Arayess, Nienke Knockaert, Bjorn Winkens, Judith W. Lubrecht, Marjoke Verweij, Anita C. E. Vreugdenhil

**Affiliations:** 1Centre for Overweight Adolescent and Children’s Healthcare (COACH), Maastricht University Medical Centre+, 6229 HX Maastricht, The Netherlands; lisanne.arayess@mumc.nl (L.A.); n.knockaert@student.maastrichtuniversity.nl (N.K.); judith.lubrecht@mumc.nl (J.W.L.); 2School of Nutrition and Translational Research (NUTRIM), Maastricht University Medical Centre+, 6229 ER Maastricht, The Netherlands; 3Department of Paediatrics, Maastricht University Medical Centre+, 6229 HX Maastricht, The Netherlands; 4Department of Paediatrics, VieCuri Hospital, 5912 BL Venlo, The Netherlands; mverweij@viecuri.nl; 5Department of Methodology and Statistics, Maastricht University Medical Centre+, 6229 HX Maastricht, The Netherlands; bjorn.winkens@maastrichtuniversity.nl; 6Care and Public Health Research Institute (CAPHRI), Maastricht University Medical Centre+, 6229 ER Maastricht, The Netherlands

**Keywords:** childhood obesity, COVID-19, pandemic, personalised lifestyle intervention, overweight, obesity, BMI

## Abstract

Background: Early research showed weight gain in children during the COVID-19 pandemic. Objective: To compare changes in BMI z-score of children with overweight and obesity in a personalised lifestyle intervention before and during the pandemic. Methods: Changes in BMI z-score half a year (T6) and twelve months (T12) after the first lockdown were included for 71 children in the ‘2020 during COVID’ group and compared to 48 children in the ‘2019 before COVID’ group, using a marginal model for repeated measures (model 1). Model 2 corrected for lifestyle intervention characteristics, and model 3 corrected additionally for family characteristics. Results: The mean difference in BMI z-score change was significantly different at T12 (+0.07 in 2020 versus −0.09 in 2019, *p* = 0.022). Model 3 showed significant differences in BMI z-score change at both T6 (+0.15, *p* = 0.024) and T12 (+0.18, *p* = 0.016). This model also defined ‘having a mother with obesity’ (+0.13, *p* = 0.019) and the frequency of no-show consultations (+0.41 per missed consultation per month, *p* = 0.025) as related factors. Conclusions: Lifestyle intervention in children with overweight and obesity is less successful in decreasing BMI z-score during the COVID-pandemic. Identified risk factors for less success could contribute to identifying children with higher risks for, and possibly prevent, BMI z-score increase.

## 1. Introduction

In 2020, the world was confronted with a pandemic caused by the SARS-CoV-2 virus. Several countries had to take governmental measures to cope with the consequences of the virus, such as national lockdowns. In the Netherlands, the first national lockdown started on 15 March 2020 and included several important measures for children, including school and sports club closures and the advice to stay at home as much as possible [1].

Although the SARS-CoV-2 virus itself seems to have a less severe pattern in children, it is assumed that the changing circumstances in daily life have led to drastic changes in the lifestyle of children [2]. Previous research showed lifestyle changes towards more unhealthy behaviour during the pandemic in both children and adults [3,4,5]. In the early phase of the pandemic, research on short term weight development was published, showing weight gain in children [6,7,8].

Children with overweight especially seem to be a risk subgroup for weight gain. Before the pandemic, studies showed the vulnerability of these children in comparable periods in terms of changing lifestyle patterns, such as school closure during summer holidays [9,10,11]. This is worrisome since childhood overweight and obesity is associated with serious health consequences, physical as well psychological, in both childhood and adulthood [12,13,14,15,16]. Childhood obesity also leads to problematic social and economic consequences, such as increasing health costs in the future [17,18,19].

Weight gain is likely to remain a persistent problem during this pandemic for children, especially for those with overweight, since the pandemic is still a threat for many countries, with corresponding governmental measures such as lockdowns. However, little is known about the long-term consequences of these lifestyle behaviours and weight changes early in the pandemic and which subgroups are at risk. Additionally, the impact of the changed circumstances on existing lifestyle interventions for children due to the pandemic is unclear.

Therefore, the aim of this study is to determine the change in BMI z-score of children with overweight and obesity in a personalised lifestyle intervention six and twelve months after the start of the first lockdown due to the COVID-19 pandemic and compare this to the same period one year earlier.

## 2. Methods

### 2.1. Setting

Data of participants were collected from the Centre for Overweight Adolescent and Children’s Healthcare (COACH) at the Maastricht University Medical Centre (MUMC+) and VieCuri Medical Centre in Venlo, the Netherlands. COACH is an expertise centre for children with overweight and (severe) obesity for both clinical evaluation and treatment, by providing a family-based, interdisciplinary, tailored lifestyle intervention. The setting and design of this intervention pre-pandemic are described in detail elsewhere [20].

Data of Maastricht were obtained within local regulations of the hospital and registered at the Ethics Committee of the Maastricht University Medical Centre (METC 2022-3105). The Ethics Committee stated that this research did not fall under the scope of the Medical Research Involving Human Subjects Act (WMO) and therefore no ethical approval and informed consent were needed. Data of Venlo was collected within the “Kijk op Overgewicht study” (METC 13-4-130, Clinicaltrial.gov (NCT02091544)). All parents and/or children in Venlo gave written informed consent for this study.

### 2.2. Participants

Participants were included in either the 2020 group, with measurements during the COVID-19 pandemic, or in the 2019 group, used as a control group.

Participants were included in the 2020 group if they had a measurement at baseline (T0) and at least one measurement after about six months (T6) and twelve months (T12). The baseline (T0) measurements were obtained between 1 January 2020 and 15 March 2020. For T6, this period was from 15 August 2020 to 15 October 2020, and for T12, from 1 January 2021 to 15 April 2021.

For 2019, these three periods ran from 1 January 2019 to 15 March 2019 (baseline, T0), from 15 August 2019 to 15 October 2019 (T6) and from 1 January 2020 to 15 March 2020 (T12). Since the lockdown started on 15 March 2020, no measurements were obtained between 15 March 2020 and 15 April 2020.

Children that participated in the long-term intervention during both 2019 and 2020, and therefore had anthropometric measurements in both years (*N* = 20), were randomly distributed between the 2019 and 2020 groups to avoid overlap and with the intention to create two independent groups.

To have a representative cohort for children with overweight in the school age, children younger than 4 years and older than 18 years at T0 or with a normal weight at T0, were excluded.

Since COACH is an ongoing, long-term family intervention, children could be in different phases of the intervention (waiting list, intake phase, diagnostic phase, active intervention or relapse prevention phase) at T0. All children with anthropometric measurements in the corresponding periods were included, regardless of the length or intensity of the lifestyle intervention.

### 2.3. Lockdown Due to the COVID-19 Pandemic in the Netherlands

The first lockdown in the Netherlands was characterised by several measures, including but not limited to school closures of both primary and high school (online education), closure of restaurants and sports clubs, cancellation of large gatherings and advice to stay at home if you have COVID-19-related symptoms, to work from home and to avoid large gatherings [21]. Schools were fully re-opened in August 2020 [22]. In December 2020, a second lockdown was announced, including school closure (online education) until February 2021 for primary schools and March 2021 for high schools [23].

### 2.4. Study Measurements

Anthropometric data (height and body weight) and data on child- and parental characteristics at the start of the intervention (sex, age, ethnicity, BMI and IOTF status of mother and father) were extracted from the medical record. 

Measurements for the height and weight of the children were obtained by a healthcare professional following the Dutch guidelines for measuring weight and height of children [24]. Most of the measurements were performed in the COACH outpatient clinic using an electric scale (Seca© 877, Seca, Hamburg, Germany) to the nearest 0.1 kg. Standing height was measured using a portable stadiometer (Seca© 213 stadiometer, Seca, Hamburg, Germany). A minority of the measurements were collected through the medical record via referrals of general youth health doctors or appointments at other disciplines of the Paediatric department of the Medical University Hospital Maastricht. 

Self-reported home measurements during remote visits (by phone or video conference) were excluded. 

BMI score (weight [kg]/height [m]^2^) and the BMI z-score were calculated in the Growth Viewer of the patient file, according to the reference data of the TNO Growth Calculator [25,26]. Weight classification was based on the International Obesity Task Force (IOTF criteria) classification system [27]. The change from baseline in BMI z-score was calculated for T6 and T12. 

### 2.5. Lifestyle Intervention Determinants

Due to the national governmental measures because of the COVID-19 pandemic, outpatient hospital visits were not possible during the first lockdown in the Netherlands. Remote consultations via phone or video conference were scheduled to stay in contact with the patients and their families. Additionally, since 15 March 2020, appointments needed to be cancelled or rescheduled when the patient or somebody in the household has symptoms related to COVID-19. The number of the different types of consultations, namely physical visit at the outpatient clinic, remote consultation (video conference or phone) or no-show (consultations that were cancelled by the patient or the family, or when the patient did not show up at the pre-arranged consultation), were retrieved from the patient files. The total number of consultations per month was calculated as the sum of the physical visits and remote consultations, divided by the time in months between T0 and the measurement moment.

### 2.6. Statistical Analysis

Data are presented as mean (SD) or the number of children (%). Independent-samples *t*-test or chi-square tests were used to examine differences in numerical and categorical characteristics between 2019 and 2020, respectively. 

Three models were created and applied to the study population. To assess the group effect (2020 versus 2019) on change in BMI z-score from baseline after 6 and 12 months, we used a marginal model for repeated measures with group, time (6 or 12 months), the interaction between group and time, and centre as fixed factors and an unstructured covariance structure for repeated measures (model 1). To get insight in the effects for the separate subgroups of children that had an increase in their BMI z-score and children that had a stabilisation or decrease in BMI z-score, this model was re-applied to those subgroups separately as a post hoc analysis. 

To account for the potential effects of lifestyle intervention factors (number of both physical visits to the outpatient clinic and offline consultations per month, as well the number of no-show consultations) on the change in BMI z-score, we added these factors to the fixed part of model 1 (model 2). As a sensitivity analysis, baseline family characteristics (ethnicity, IOTF classification at T0, length in intervention, educational level of parents, age or having a mother or father with obesity) were separately added to model 2 to see which characteristics contributed significantly to the model. All characteristics that were significantly contributing to the model were included in the final model (model 3). Estimates of fixed effects together with their 95% confidence intervals and *p*-values are presented. Two-sided *p*-values ≤ 0.05 were considered statistically significant.

Statistical analyses were performed using IBM SPSS Statistics for Windows version 25 (Armonk, NY, USA).

## 3. Results

### 3.1. Baseline Characteristics

Seventy-one children had a measurement of their height and weight in the three months prior to the start of the lockdown in the Netherlands (15 March 2020) and were included in the 2020 group, while 48 children were included in the control group (2019). (Figure 1) Baseline characteristics regarding weight status, family factors and length in the intervention were similar between the groups (Table 1).

### 3.2. BMI z-Score Change for Children with Overweight and (Severe) Obesity on the Mid-Long Term after the Start of the COVID-19 Pandemic

In 2020, 30 (51.7%) children had an increase in BMI z-score at T6 when compared to T0, while there were 19 children (48.7%) with an increase in the control group of 2019. 

At T12, 29 (60.4%) and 18 (58.1%) children had an increase of the BMI z-score in 2020 and 2019, respectively. The change in BMI z-score at T6 and T12 in the 2020 group shows an increasing trend, compared to a decreasing trend in 2019 (Figure 2).

Based on model 1, the mean difference in BMI z-score change was not significant at T6 (+0.05 in 2020 versus −0.05 in 2019, difference = +0.10, 95% CI −0.01, +0.21, *p* = 0.061), while it was significant at T12 (+0.07 in 2020 versus −0.09 in 2019, difference = +0.16, 95% CI 0.02, 0.30, *p* = 0.022).

The same model was re-applied for the different subgroups of children that had an increase in their BMI z-score and children that had a decrease in BMI z-score. This showed significant differences at both T6 and T12 when 2020 was compared to 2019 for children with a BMI z-score increase (mean difference at T6 = 0.10, 95% CI 0.00, 0.21, *p* = 0.047 and mean difference at T12 0.22, 95% CI 0.11, 0.34, *p* < 0.001). It also showed a significant difference at T6 for children with a BMI z-score decrease when 2020 was compared to 2019 (mean difference 0.11, 95% CI 0.03, 0.20, *p*-value = 0.010) (Table 2).

### 3.3. Lifestyle Intervention Changes during the COVID-19 Pandemic

When characteristics of the lifestyle intervention between the group in 2020 and 2019 were compared, no significant differences in total consultations per month or no-show consultations were observed. However, significant differences were observed in the number of outpatient clinic visits and remote consultations when 2020 and 2019 were compared (Figure 3).

According to model 2, i.e., after correction for the lifestyle intervention characteristics, the mean change in BMI z-score was significantly higher in 2020 when compared to 2019 at both T6 and T12 (see Table 3).

A significant contributing lifestyle factor in this model was the frequency of no-show consultations per month. With every missed consultation per month, the change in BMI z-score at both time points increased by +0.43 (95% CI 0.07, 0.80, *p* = 0.021).

### 3.4. Model 3: Creating a Model to Identify Family Characteristics Influencing BMI z-Score Change

The only family characteristic that added significantly to model 2, and was therefore included in model 3, was having a mother with obesity (*p* = 0.019).

This model also showed that the corrected mean differences in change of BMI z-score were significant, both at T6 (+0.07 in 2020 versus −0.08 in 2019, difference = 0.15, 95% CI 0.02, 0.27, *p* = 0.024) and T12 (+0.08 in 2020 versus −0.10 in 2019, difference = 0.18, 95% CI 0.03, 0.32, *p* = 0.016). Additionally, the frequency of no-show consultations remained a significant contributor (*p* = 0.025).

## 4. Discussion

To the best of our knowledge, this is the first study that shows the weight gain in children with overweight and (severe) obesity on the mid-long term, approximately half a year and one year after the start of the COVID-19 pandemic. This study adds knowledge to previous studies on children of the general population by showing the drastic changes in BMI z-score in children with overweight and (severe) obesity that are already participating in a lifestyle intervention. Under pre-pandemic circumstances in 2019, the mean BMI z-score change over the total cohort in the lifestyle intervention is decreasing at both time points, in line with previously described positive effects of the COACH personalised lifestyle intervention [20,28]. Unfortunately, during the lifestyle intervention in 2020, the mean BMI z-score change for children increased. Even after correction for several determinants related to the lifestyle intervention and the family, such as frequency of consultations, our models showed that the difference in change of BMI z-score was significantly higher when 2020 was compared to 2019.

Furthermore, the mean BMI z-score increase for children with weight gain in the lifestyle intervention was significantly larger in 2020 when compared to 2019. Additionally, the success of the lifestyle intervention six months after the lockdown for the subgroup of children that had a decrease or stabilisation of the BMI z-score was less, since our study found a smaller mean BMI z-score decrease at T6 in 2020 when compared to 2019. These results should be considered alarming.

It should be noted that the children in this study were participating in a personalised lifestyle intervention. In general, the main goal of this lifestyle intervention is a decrease in BMI z-score of 0.15 SDS, based on the relationship with cardiovascular health outcomes, although this may vary depending on individual characteristics such as age, severity of the overweight and phase in the intervention [29]. Since overweight is known to be associated with several weight-related comorbidities, even at a younger age, and an increasing BMI z-score is continuously associated with cardiovascular complications such as high blood pressure and dyslipidaemia, it is advisable to keep track of the weight status of children with overweight and obesity [13,30]. However, our study results suggest this is even more important during a pandemic, since even the help of a lifestyle intervention in an expertise centre was not sufficient for stabilising or decreasing the mean BMI z-score of the cohort. Furthermore, we do know that lifestyle interventions for children with overweight in general are efficacious in the treatment of paediatric overweight when compared to children who do not receive guidance [31]. A previous study from our research group showed that parents of children with overweight or obesity more frequently reported perceived weight gain during the lockdown, when compared to parents of children with normal weight [7]. Therefore, the increase in BMI z-score in our 2020 group could be even larger for the youth with overweight that did not receive guidance, since the lifestyle intervention aimed to create guidance and an incentive for healthy behaviour in this study group during the pandemic. To cope with the changed daily lives of children with overweight during the COVID-19 pandemic and possibly other crisis periods, lifestyle interventions should be attentive to the impact of the COVID-19 pandemic and its effects and consider adaptations. Therefore, this study forms the basis for further studies to obtain more insight in contributing determinants.

Our results highlight the importance to take family-related determinants into account in a lifestyle intervention, as children of mothers with obesity were especially at risk for an increase in BMI z-score. It is known that parental obesity and lifestyle influence the weight status of children [32,33]. However, it remains unclear to what extent a pandemic influences these family-related associations. A qualitative study found that parents were concerned that their child with overweight has more access to unhealthy food and that the child is more frequently in surroundings where overweight is normalised [34]. Additionally, other mental health stressors were mentioned in the literature, possibly influencing lifestyle behaviour [35]. To obtain more in-depth data on how the lifestyle in families exactly changed for children with overweight during COVID-19, and what changes in lifestyle behaviours are long-lasting and influencing lifestyle interventions, more long term and qualitative data is needed.

Besides the previously mentioned influence of the weight status of the mother, most of the family determinants in this study did not seem to influence the BMI z-score change of children. Possibly, data on parental variables (weight status or educational level) could be outdated or less accurate for children that are in the lifestyle intervention for a longer period, since data of parents is self-reported at the intake of the lifestyle intervention. Specifically, anthropometric data of fathers were not complete for all children, leading to higher inaccuracy for these variables. This is considered a limitation of this study.

The study provides new insights into the characteristics of a lifestyle intervention during a pandemic, showing the total number of consultations per child did not differ significantly between 2019 and 2020 although a shift to more remote consultations was observed. In addition, it indicates that the number of physical and remote consultations did not have a significant influence on the BMI z-score change. These results are in line with the goal of the personalised intervention, that determines the frequency of contact that is adapted by individual and family needs and possibilities. However, the number of no-show appointments was significantly, positively associated with the change in BMI z-score. Thus, missing appointments most likely will lead to an increase in BMI z-score. A previous Dutch study assessing barriers and facilitators in the adherence to a lifestyle group intervention for children with obesity, describes motivation, satisfaction and means (such as accessibility and time) as barriers [36]. Specifically for 2020, it is hypothesised that the families with high no-shows had less access to online possibilities, such as remote consultations and online lifestyle activities, or are less motivated to make changes in the lifestyle of the child, because of other priorities or less awareness of overweight. Since anxiety during the pandemic in children with overweight is described previously, it is also possible that families with high no-shows were more afraid of COVID-19 and were also avoiding other contacts and activities besides the intervention, and therefore changed their daily lifestyle more [37]. Regardless of the reasons for missing the appointments, this study illustrates the importance of adherence to a tailored intervention. Since personalisation of a lifestyle intervention is seen as a facilitator in adherence, it is advisable to consider children with high no-show rates after the start of the pandemic as a risk group for weight gain and to adapt the interventions to the needs of these families [36,38].

A previous study of our group, conducted directly after the end of the first lockdown, showed that in children with obesity that participated longer than 1 year in the lifestyle intervention, no significant increase in BMI z-score was shown [7]. In contradiction, the current study in this cohort did not see any influence of the length of the intervention on the BMI z-score change. This difference in results is an important finding, that requires deeper analysis. An explanation could be that the length of the pandemic, and therefore the duration of the changed circumstances, could potentially have a stronger negative impact on the lifestyle and weight, than the preparation that was learned at the lifestyle intervention before the pandemic. Given the threat of new COVID-19 virus mutations with possible new corresponding governmental measures influencing daily lifestyle, it is of great importance to keep track of the weight status of children with overweight and obesity in the upcoming period [39,40].

## 5. Conclusions

Overall, the current study shows the alarming results of weight gain one year after the start of the COVID-19 pandemic in a unique cohort of children within a lifestyle intervention that already suffer from overweight or (severe) obesity. This should be a concern for not only children, parents and health care professionals, but also governments, as it threatens both the current and the future public health, with potential risks for the health of individual children, and financial and social long-term effects. It also shows that, although lifestyle interventions were forced to make changes in their programs due to regulations, such as offering (more) remote consultations, possibly more adaptations should be made to previously successful lifestyle interventions to cope with the changed circumstances of a pandemic. The outcomes of this study clearly show the subgroups that are more at risk for weight gain within a lifestyle intervention, such as children with a mother with obesity and children with high no-show rates to the lifestyle intervention. Thus, health care professionals working with children with overweight and obesity should especially focus on these two subgroups during and post-pandemic. These results offer the first opportunities to tailor lifestyle interventions for these risk groups during and after the current pandemic and future crisis periods.

## Figures and Tables

**Figure 1 nutrients-14-01942-f001:**
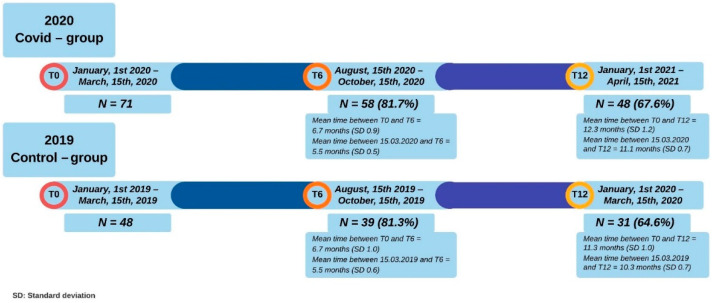
Timeline and inclusion at the several measurement moments for the 2020 COVID-19 group and the 2019 control group.

**Figure 2 nutrients-14-01942-f002:**
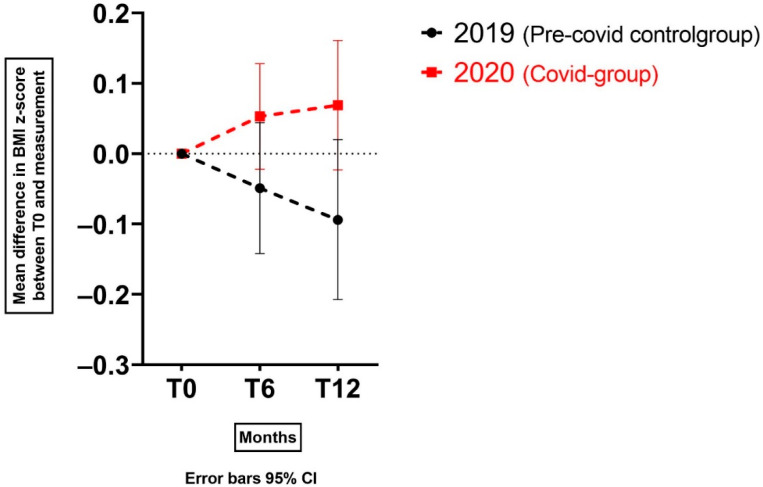
Change in BMI z-score for children with overweight and (severe) obesity at T6 and T12 in 2019 and 2020.

**Figure 3 nutrients-14-01942-f003:**
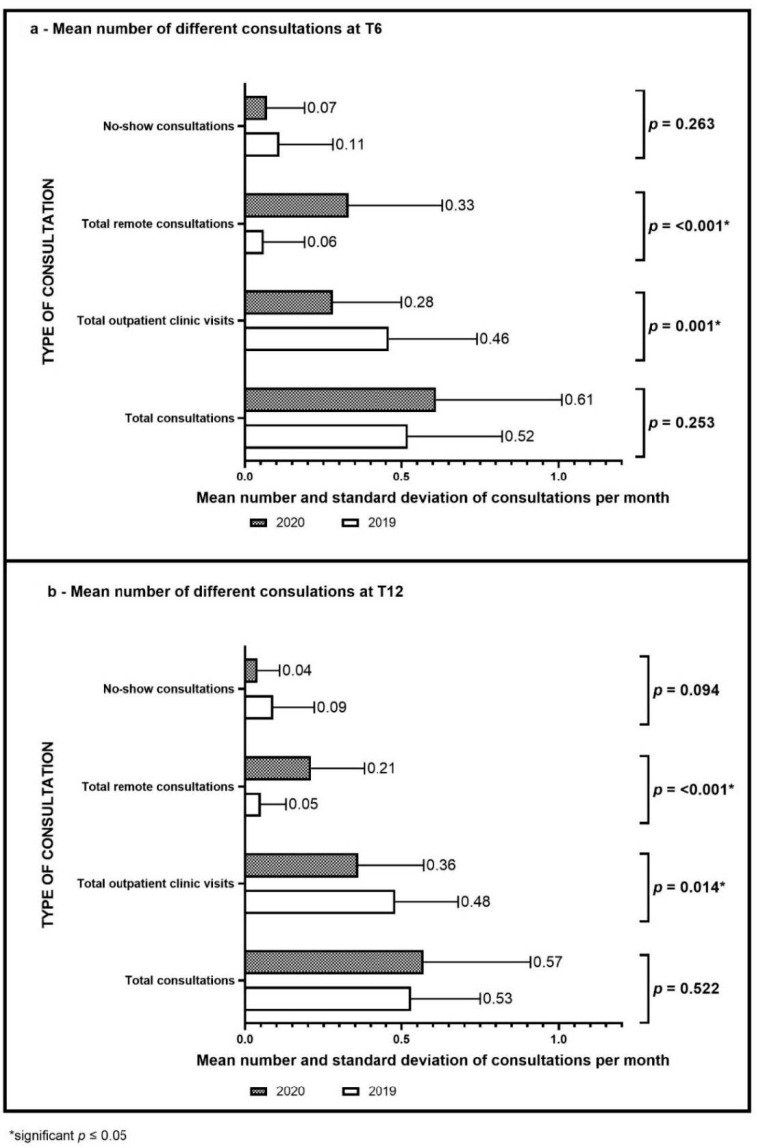
Characteristics of the lifestyle intervention: different consultations at T6 (**a**) and T12 (**b**) in 2020 and 2019.

**Table 1 nutrients-14-01942-t001:** Baseline characteristics.

	2020 (COVID-19 Group)	2019 (Control Group)	*p*-Value
	N = 71	N = 48	
**Age, mean (SD); years**	12.6 (3.1)	11.7 (2.5)	0.094
**Gender, % female (*N*)**	49.3 (*N* = 35)	52.1 (*N* = 25)	0.765
**BMI score (SD); kg/m^g^**	28.59 (5.95)	27.56 (3.46)	0.237
**BMI z-score (SD)**	3.09 (0.70)	3.11 (0.60)	0.888
**IOTF at T0**			0.744
** *Overweight, % (N)* **	33.8 (*N* = 24)	29.2 (*N* = 14)	
** *Obesity, % (N)* **	40.8 (*N* = 29)	47.9 (*N* = 23)	
** *Severe obesity, % (N)* **	25.4 (*N* = 18)	22.9 (*N* = 11)	
**Months in intervention at T0 (SD) mean**	13.1 (18.3)	15.0 (18.7)	0.578
**<1 year since start intervention at T0, % (*N*)**	59.2 (*N* = 42)	64.6 (*N* = 31)	
**>1 year since start intervention at T0, % (*N*)**	40.8 (*N* = 29)	35.4 (*N* = 17)	0.551
**Ethnicity**			0.856
** *Dutch, % (N)* **	62.9 (*N* = 39) #	64.6 (*N* = 31)	
** *Migration background, % (N)* **	37.1 (*N* = 23) #	35.4 (*N* = 17)	
** Parental factors **			
**BMI mother, mean (SD); kg/m^g^ **	28.41 (4.99) ^	30.31 (6.08)^	0.068
**BMI father, mean (SD); kg/m^g^**	29.67 (4.91) ^^	29.48 (4.82) ^^	0.845
**Having a mother with obesity, % (N)**	40.6 (*N* = 28) ^	46.8 (*N* = 22) ^	0.506
**Having a father with obesity, % (N)**	36.7 (*N* = 22) ^^	42.5% (*N* = 17) ^^	0.558
**Having** **≥** **1 parent with obesity, % (N)**	63.5 (*N* = 40) ^^^	65.1 (*N* = 28) ^^^	0.864
**Educational level mother**			
*Low, % (N)*	38.9 (*N* = 21) ◊	34.9 (*N* = 15) ◊	0.16
*Medium, % (N)*	40.7 (*N* = 22) ◊	27.9 (*N* = 12) ◊	
*High, % (N)*	20.4 (*N* = 11) ◊	37.2 (*N* = 16) ◊	
**Educational level father**			
*Low, % (N)*	32.0 (*N* = 16) ◊◊	34.1 (*N* = 14) ◊◊	0.89
*Medium, % (N)*	34.0 (*N* = 17) ◊◊	36.6 (*N* = 15) ◊◊	
*High, % (N)*	34.0 (*N* = 17) ◊◊	29.3 (*N* = 12) ◊◊	

# Data available for *N* = 62 in 2020. ^ Data available for *N* = 69 in 2020, *N* = 47 in 2019. ^^ Data available for *N* = 60 in 2020, *N* = 40 in 2019. ^^^ Data available for *N* = 63 in 2020, *N* = 43 in 2019. ◊ Data available for *N* = 54 in 2020, *N* = 43 in 2019. ◊◊ Data available for *N* = 50 in 2020, *N* = 41 in 2019.

**Table 2 nutrients-14-01942-t002:** Mean change for subgroups that had an increase or decrease/stabilisation in the BMI z-score at T6 and T12 in 2020, when compared to 2019.

	T6		*p*-Value ^	T12		*p*-Value ^
	2020	2019		2020	2019	
BMI z-score increase, mean (SD) change for subgroup	0.24 (0.03) *N* = 30	0.14 (0.04) *N* = 19	0.047 *	0.32 (0.04) *N* = 29	0.10 (0.05) *N* = 18	<0.001 *
BMI z-score decrease or stabilisation, mean (SD) change for subgroup	−0.16 (0.03) *N* = 28	−0.27 (0.04) *N* = 20	0.010 *	−0.23 (0.05) *N* = 19	−0.34 (0.06) *N* = 13	0.178

* Statistically significant, *p* ≤ 0.05. ^ based on model 1, which means that the differences in mean change from baseline scores (at T6 and T12) are corrected for centre, measurement moment, year*measurement moment.

**Table 3 nutrients-14-01942-t003:** Estimated fixed effects in the model for BMI z-score difference for children with overweight and (severe) obesity.

	Model 2				Model 3			
Parameter	Estimate	95% CI		*p*-Value	Estimate	95% CI		*p*-Value
		Lower Bound	Upper Bound			Lower Bound	Upper Bound	
Year [2020 versus 2019] at T6	+0.14	0.01	0.27	0.033 *	+0.15	0.02	0.27	0.024 *
Year [2020 versus 2019] at T12	+0.18	0.04	0.33	0.014 *	+0.18	0.03	0.32	0.016 *
Centre [Maastricht]	+0.04	−0.10	0.18	0.602	+0.05	−0.09	0.19	0.472
Contact moments outpatient clinic/month	+0.05	−0.15	0.25	0.635	+0.05	−0.15	0.25	0.609
Remote contact moments/month	−0.03	−0.24	0.17	0.752	−0.00	−0.21	0.20	0.993
No-show appointments per month	+0.43	0.07	0.80	0.021 *	+0.41	0.05	0.77	0.025 *
>=1 Year in intervention at baseline	−0.02	−0.14	0.09	0.680	−0.02	−0.13	0.10	0.772
Having a mother with obesity					+0.13	0.02	0.23	0.019 *

* Significant, *p* ≤ 0.05.

## Data Availability

The data presented in this study are available on request from the corresponding author. The data are not publicly available due to privacy and ethical rules.

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
