# Peer review of "The Side-Effects of the COVID-19 Pandemic: Increased BMI z-Score in Children with Overweight and Obesity in a Personalised Lifestyle Intervention One Year after the Start of the Pandemic in The Netherlands"

_nutrients, 2022, doi:10.3390/nu14091942_

Round 1

Reviewer 1 Report

In this manuscript, authors reported the possible impacts of pandamic lockdown on BMI z-score in overwight/obese children with lifestyle interventions. It is of high interests to readers. However, several concerns need to be addressed. 

a. Table 1. The representative values for Variables "Having a father with obesity, % (N)" and "Having at least one parent with obesity, % (N)" are not placed properly. The n=28 and 22 for Variables "Having a mother with obesity, % (N)" and "Having a father with obesity, % (N)" in 2020, respectively. However, there are only n=40 for " having at least one parent with obesity".  Should the n number be at least 50 since there are 28 children haveing a father with obesity and 22 having a mother with obesity? Please clarify the n number. 

b. Table 3. why the variables are calculated as "total amout per month" rather reporting the total amount of conculatioins or outpatient clinic visits? 

c. Line 220-221, authors stated that "the group effect (2020 versus 2019) was significant at T6 and T12". Howver, for readers who are interested in this study but not familiar with the statistical model/outcomes, it will be difficult to interpret the data. For example, what is this significant group effect mean? Is the BMI z-score significantlyl increased or decreased at T6 in year 2020 vs 2019? 

Reviewer 2 Report

The topic choose by the author is new. But the representation of the article is not great to express huge interest to the reader. The introduction and discuss should be more compact and crisp in terms of writing. Instead of putting almost every data (except one) in table, author should consider to form a bar diagram or something else for table3. These minor changes will flourish the quality of the article. 
